# Test Your Memory (TYM) and Test Your Memory for Mild Cognitive Impairment (TYM-MCI): A Review and Update Including Results of Using the TYM Test in a General Neurology Clinic and Using a Telephone Version of the TYM Test

**DOI:** 10.3390/diagnostics9030116

**Published:** 2019-09-08

**Authors:** Jeremy M. Brown, Julie Wiggins, Kate Dawson, Timothy Rittman, James B. Rowe

**Affiliations:** 1Department of Neurology, Box 83, Addenbrooke’s Hospital, Hills Road, Cambridge CB2 2QQ, UK; timothy.rittman@nhs.net (T.R.);; 2Department of Clinical Neurosciences, University of Cambridge, Cambridge CB2 0SZ, UK; jkw41@medschl.cam.ac.uk (J.W.); ced35@cam.ac.uk (K.D.)

**Keywords:** dementia, TYM, TYM-MCI, Alzheimer’s

## Abstract

This paper summarises the current status of two novel short cognitive tests (SCT), known as Test Your Memory (TYM) and Test Your Memory for Mild Cognitive Impairment (TYM-MCI). The history of and recent research on the TYM and TYM-MCI are summarised in applications for Alzheimer’s and non-Alzheimer’s dementia and mild cognitive impairment. The TYM test can be used in a general neurology clinic and can help distinguish patients with Alzheimer’s disease (AD) from those with no neurological cause for their memory complaints. An adapted tele-TYM test administered by telephone to patients produces scores which correlate strongly with the clinic-administered Addenbrookes Cognitive Examination revised (ACE-R) test and can identify patients with dementia. Patients with AD decline on the TYM test at a rate of 3.6–4.1 points/year.

## 1. Introduction

NICE has recommended the use of the TYM test in non-specialised settings, and this paper is intended to facilitate its use and understanding [1].

### 1.1. The TYM Test

The original TYM test is a series of 10 cognitive tasks printed on both sides of a thin sheet of card that is filled in by a patient under minimal supervision by a non-specialist healthcare worker [2]. How much help the patient needs is used as an 11th scored “task” of the test. The TYM test can be viewed and downloaded free from the website tymtest.com. The vast majority of patients needs little help completing the TYM test and this allows the supervising health worker to continue to perform other tasks at the same time, e.g., welcoming patients to the clinic, booking patients in and out, or filling in a scan form. Scoring the TYM test takes approximately two minutes. The TYM test therefore takes a minimal amount of medical time to administer. Much of the administration and scoring of the test is intuitive, meaning staff need minimal training to supervise and score the test. JMB (Jeremy M. Brown) has successfully taught many nurses at the Queen Elizabeth Hospital, King’s Lynn, and elsewhere to supervise the test at the start of the clinic, even if they have no experience of memory clinics. It is possible to obtain valid results with patients’ relatives rather than health professionals supervising the test [3]. A sheet is available from the website tymtest.com to aid scoring and more detailed scoring sheets are available from JMB.

The 11 tasks cover a wide range of cognitive skills similar to those tested by the Addenbrookes Cognitive Examinations (ACE) [4,5,6] and in all our studies the TYM score is very strongly correlated to the ACE-R (Addenbrookes Cognitive Examination Revised) score (with high Pearson co-efficient). The ACE-R and ACE-III are the gold standard short cognitive tests in use in memory clinics, and are widely used throughout the world. The TYM test is scored out of 50 and the ACE-R and ACE-III are scored out of 100; doubling the TYM score will give a good estimate of the ACE-R or ACE-III score.

### 1.2. Domains of the TYM Test

Orientation to time and person (orientation)—scored out of 10Copying a sentence (copying)—scored out of 2Recall of facts (facts)—scored out of 3Arithmetic (sums)—scored out of 4Animal Fluencies (fluencies)—scored out of 4Similarities—scored out of 4Naming—scored out of 5Spotting the letter W (visuospatial 1)—scored out of 3Completing a clock (visuospatial 2)—scored out of 4Recall of the sentence (recall)—scored out of 6Help needed to complete the test (help)—scored out of 5

## 2. History

The TYM test was invented and developed by the author from 2007 to 2009 and the original validation paper was published in the BMJ (British Medical Journal) in 2009 [2]. This showed that the TYM test detected the majority of patients with Alzheimer’s disease. There was considerable initial publicity and interest and almost immediately a large number of validation projects began throughout the world. Many of these have now been published in peer-reviewed journals. A table giving a review of these papers was included in a chapter in 2015, and since then more validation studies have been published. Table 1 includes a summary of the results of all of these studies, which include over 4000 individuals. The protocols for the different studies vary and the populations studied were very different, so there is limited value in comparing sensitivities and specificities.

All of the studies published are positive, demonstrating that the TYM test is a good short cognitive test for the detection of dementia in a wide range of cultures, in different languages, and using different alphabets. The test was developed in the United Kingdom and does show a cultural bias. This has necessitated some country-specific modifications of the TYM test, which are usually minor, e.g., changing important national dates or the letter for animal fluencies. The sentence about “stout shoes” is very British and the alternative sentence “Great cooks always bake chocolate cakes” has been used in several validations. It is important that the sentence is not a well-known phrase and it needs to be counter-intuitive to avoid a patient guessing the end by remembering the first couple of words. The sentence should not be directly translated, as to do so may undermine its critical features for cognitive assessment. Information on how to adapt the TYM test for different languages is available from JMB.

The tymtest.com website was launched in 2010. Health professionals can view the test and download the TYM test from the site free of charge for healthcare and research purposes. They can also download scoring instructions and advice on how to administer the test. The website continues to be widely used and the TYM test has now been downloaded more than 20,500 times from the website.

The website is currently being upgraded and videos will be added to aid the use and standardisation of the TYM test. The TYM-MCI will also be available from the site. 

### 2.1. TYM Tests for Non-Alzheimer’s Dementia

Further studies on the TYM test by the author have also confirmed the initial findings [27]. Recently the author published a validation study of the TYM test in non-Alzheimer’s dementia [28]. These included the behavioural variant of frontotemporal dementia, semantic dementia, vascular dementia, progressive non-fluent aphasia, corticobasal syndrome, progressive supranuclear palsy and atypical focal presentations of Alzheimer’s disease (posterior cortical atrophy and logopaenic aphasia). 

The TYM test was a useful test in detecting each of these different dementia types. The standard TYM test (cut off 42) detected 80% of non-Alzheimer’s or atypical Alzheimer’s dementia, whereas the ACE-R (cut off 82) detected 69% of cases and the MMSE (Mini-mental state examination) (cut off 23) detected only 27%. When grouping all of the different diseases together, the TYM test had a positive predictive value of 0.80 and a negative predictive value of 0.84, with an area under the ROC curve of 0.89, similar to its accuracy at detection of typical Alzheimer’s disease. The pattern of scoring on the TYM test differed between conditions, and the pattern of deficits can be useful in supporting a clinical diagnosis, particularly for semantic dementia and posterior cortical atrophy.

### 2.2. The TYM-MCI

The TYM test has been shown to be useful in the detection of both Alzheimer’s and non-Alzheimer’s dementia. Empirically it is unlikely to be as useful in detecting amnestic mild cognitive impairment, an important role in the memory clinic, as it has 9 points for memory, and so a patient with single cognitive domain deficits in memory may need to score just 2/9 in order for them to reach the threshold of 43/50. 

The detection of early or mild clinical cases of aMCI (amnestic Mild Cognitive Impairment) or early symptomatic AD is important, particularly as disease modifying treatments are developed. JMB recognised the need for a short cognitive test aimed at better sensitivity to aMCI and this led to the TYM-MCI test. During its development, the TYM-MCI was known first as the Hard TYM (H-TYM) or the Tricky TYM. The current and more formal name TYM-MCI is the most apt name, describing exactly what it does—it detects mild cognitive impairment. 

The TYM-MCI is the first short cognitive test to examine both visual and verbal episodic memory. It looks rather similar to the TYM test and will be available shortly on the website tymtest.com. However, the usage of the TYM-MCI is very different from the TYM test—the TYM-MCI is administered by a health professional. The administration takes approximately 7 minutes. The TYM-MCI should only be used for the detection of aMCI or mild AD (unlike the TYM test, which can be used in many clinical scenarios), and the patient should have completed and performed well on a standard SCT (either TYM or other appropriate SCT) prior to attempting the TYM-MCI. It can be run immediately following the standard TYM in a clinic setting. As the diagnostic value rests on the patient’s recall of recently learnt visual and verbal information, only the recall tasks on page 2 are scored. The scoring system will be released shortly on the website tymtest.com and is available from JMB. Visual recall and verbal recall are both scored out of 15, giving a maximum total score of 30. 

Two validation studies of the TYM-MCI have been reported [29,30] and both showed very striking results. The initial validation compared the TYM-MCI scores in patients with mild AD and normal controls [29]. This kind of validation study, a proof of principle, sets a fairly low bar, which the TYM-MCI nonetheless passed very easily and with very clear results. Patients with mild AD scored 6.7/30 on the TYM-MCI compared to normal controls who scored 20.4/30. The area under the ROC (receiver operating characteristic) curve was 0.99. The results were particularly impressive for recall of visual encoded information with both a modal and median score of 0/15 for patients with mild AD or aMCI compared to a mean of 10/15 for the controls. The greater deficit in visual recall may reflect preferential involvement of recall of visual information in the earliest stages of AD or may reflect that it is a more difficult sub-test of the TYM-MCI than the recall of verbally encoded information. One other H-TYM study confirmed it was a useful test [31].

The second validation study [30] was more clinically relevant, examining the usefulness of the TYM-MCI in the separation of patients with aMCI or mild AD from patients presenting to the same memory clinic with memory complaints that were felt not to have a neurological cause (subjective memory complaints). These patients could be “worried well”, depressed, or have conditions such as obstructive sleep apnoea. The study was a thorough assessment of the TYM-MCI, for example including only patients who scored in the normal range on the MMSE. 

The TYM-MCI performed very well, with a specificity of 0.79 and a sensitivity of 0.91 in distinguishing the two groups of patients. An important result of this study was that the TYM-MCI was shown to provide additional information to the ACE-R and add information to the overall clinical assessment. Three useful practical conclusions from this paper were:The combination of the TYM-MCI with the ACE-R detects virtually all cases of aMCI or mild AD.The TYM-MCI is a useful test in patients with borderline ACE-R scores. A total of 72% of patients with aMCI or mild AD who had a borderline score on the ACE-R (scoring between the 2 thresholds of 82 and 89) were detected on the TYM-MCI and 79% of patients with subjective memory complaints and a borderline ACE-R score passed the TYM-MCI.Patients initially felt to have subjective memory problems who were later re-diagnosed with organic aMCI scored poorly on their initial TYM-MCI (mean score 9.8/30).

The TYM-MCI is now used routinely in the Addenbrooke’s memory clinic for patients whom we suspect have early AD/aMCI as the cause of their memory problems but whom have scored well (>82/100) on the ACE-R.

### 2.3. TYM Testing in a Non-Specialised Clinical Settings

The TYM was designed for use in non-specialised clinics and NICE (National Institute for Health and Care Excellence) has recommended the use of the TYM test in non-specialised settings [1]. The validation research studies were undertaken in primary, secondary, and tertiary care settings. In this section, we outline the use of TYM testing in a NHS (National Health Service) Trust hospital’s general neurology clinic at Queen Elizabeth Hospital, King’s Lynn, United Kingdom. 

Research validation of the TYM test in general clinics has different endpoints to the memory clinical validation against ACE-R or ACE-III and multidisciplinary team review. In the generalist setting, there is often no definitive “memory” outcome to the consultation. However, feedback on its widespread use suggests that it does provide useful information. One study has reported on the results of TYM testing in a general neurology clinic [21], but again outcomes regarding diagnostic accuracy are difficult to assess. Here we describe our own work on the TYM test in general neurology clinics.

Materials and Methods: Between January 2013 and August 2018, a total of 938 TYM tests were administered in general neurology clinics at Queen Elizabeth Hospital, King’s Lynn, United Kingdom. The age of the patients tested ranged from 18 to 93 years of age. The average age was 65.9 years. The majority of TYM tests were performed on patients with memory complaints either as a sole problem or as part of a neurological illness, such as Parkinson’s disease, multiple sclerosis or epilepsy. The general clinic nurses supervised the TYM tests with minimal training and in addition to their usual clinic duties. There were no major problems or complaints from supervising staff or patients. 

Results: Many of the diagnoses recorded in the letters were not definitive and patients were not necessarily followed-up, so we do not present its diagnostic accuracy against a gold standard of specialist review and extensive investigations. The final diagnosis could not be determined for many patients and these individuals are not included in the analysis. These data are therefore “pragmatic” and embedded in a real-world setting. The diagnoses were clinical and not made according to international criteria, so detailed analysis is not appropriate. The data are summarized in Table 2.

Patients with a clinical diagnosis of Alzheimer’s disease had significantly lower TYM scores than those with subjective memory loss (*p* < 0.001)

### 2.4. Rates of Change in Clinical Alzheimer’s Disease, Clinical Parkinson’s Disease and Patients Felt to Have Subjective Memory Complaints

Seventeen patients with a final diagnosis of clinical Alzheimer’s disease completed the TYM test on more than one visit (follow-up 6–50 months). The rate of decline was variable between individuals and there was no clear relationship to disease severity. The mean rate of deterioration was 4.1 points/year, in keeping with a typical rate of decline of 10 points per year on the ACE-R. 

Twenty patients with clinical Parkinson’s disease had TYM tests on more than one visit (range 5–60 months). Four patients had an improvement in their score, 4 scored the same, and 12/20 (60%) showed a decline. The rate of decline was variable between individuals and there was no clear relationship to disease severity. Overall, the rate of decline in TYM scores in PD was 1.4 points/year. 

Twenty-five patients with a final baseline diagnosis of subjective memory problems completed the TYM test on more than one visit (follow-up 4–50 months). Interestingly, 18/25 showed a small improvement in their TYM score, 4 had the same score, and only 3 declined. The mean rate of change in the TYM score was plus 1.2 points/year. It seems most likely that a mild practice effect was responsible for the improvement in TYM scores. 

## 3. Conclusions

The TYM test has been used since 2013 without any complications or complaints. Patients were tested because they had memory complaints or were suspected of having cognitive problems. The major categories of patients were those suspected of having Alzheimer’s disease or other dementia, patients with subjective memory problems, Parkinson’s disease, epilepsy and mild cognitive impairment. In patients who were followed-up with, the TYM test proved very useful, as patients with AD declined at an average rate of 4.1 points per year, whereas patients with subjective complaints tended to show a small increase in TYM scores with time. Patients with Parkinson’s disease had an average score of 40.2/50 on the TYM test—below the threshold for supporting a diagnosis of dementia. PD patients did decline on the TYM test but at a much slower rate than patients with AD (1.4 points/year). The TYM test reveals memory problems in the other groups of patients seen in a General Neurology clinic but the numbers of these patients were too small to make clear conclusions.

### 3.1. The Telephone TYM or Tele-TYM

While supervised TYM testing in a clinic setting works well, there are situations where a remote assessment would be helpful, such as pre-admission clerking or follow-up. Patients may also be more relaxed and perform better at home, while it could save an outpatient appointment or allow better planning of appointments (e.g., to decide who might benefit from a pre-clinic brain scan). In principle, one could assess cognitive function over the telephone or using an internet-based consultation. Cognitive testing of patients by telephone is not straightforward but could bring advantages to both health care professionals and patients. A telephone TYM test could be used to help assess the response of a patient to a cholinesterase inhibitor and save a journey to outpatients. 

As the TYM test is filled in by the patient, it has the potential to be administered by telephone, and we investigated this in a pilot study in 2017. 

The protocol was as follows. Patients who were due to be seen in a specialised memory clinic in approximately 6 weeks were contacted by a research nurse and offered the chance to participate in the telephone TYM study. If they consented to the trial, a date about 2 weeks in advance of the clinic was agreed. They were asked to ensure that a relative or friend who was able to help them was available at that time. 

They were sent an envelope with details of the trial, including an envelope containing the TYM test, which was marked “not to be opened before nurse phones”. There was also a stamped, addressed envelope for the patient to return the completed test to the clinic nurse. 

The nurse would phone the patient at the agreed time. She would ask them to sit down in a comfortable chair at a table with their reading glasses, a pen and a helper. They were then asked to open the envelope. The nurse encouraged them to fill in the TYM test themselves as much as possible but remained on the end of the phone and provided assistance or reassurance when required. She recorded how much time and help the patient needed. She reminded them to remember the sentence at the end of page 1 and not to turn the page back. Once they finished, the nurse asked them to place the completed TYM test in the envelope and post it to her.

When the patient came to the clinic they had the usual assessment, which includes seeing an experienced nurse who would administer the ACE-R and MMSE, followed by a neurologist, imaging as required, and multidisciplinary team review. Results of the telephone TYM test were then compared to the ACE-R and MMSE results. A subset of the patients had an additional TYM test in the clinic (often a different version of the TYM test, such as the TYM-B (Test Your Memory version B). Practice effects can be seen in the TYM test if it is administered twice at short intervals. This can be only partially alleviated by using a different version of the TYM test and we normally do not repeat the TYM test at an interval of less than 3 months. 

117 patients were recruited. Of these, 16 failed to complete the TYM test or did not attend their clinic appointment. Additionally, 22 patients had diagnoses such as Parkinson’s disease, stroke or epilepsy which may or may not be associated with an organic dementia, and so were not included in the analysis. A total of 81 patients were included in the analysis—43 patients were felt to have subjective memory problems with no identified neurological cause for their memory problems after their assessment by a consultant with a special interest in cognition; 38 patients were given organic cognitive diagnoses. The breakdown of the organic cases was: mild cognitive impairment, 18; Alzheimer’s disease, 9; unspecified organic dementia, 5; semantic dementia, 2; mixed dementia, 2; dementia with Lewy bodies, 1; progressive supranuclear palsy, 1. 

Patients with an organic cause for their memory problems were significantly older than patients with subjective memory complaints. However, in this study there was no correlation between TYM score and age. Patients with organic disease scored lower in all three cognitive tests than those with subjective memory complaints but no underlying neurological disease (Table 3). There was a strong correlation between the telephone TYM score and the clinic ACE-R score in both groups (Pearson *r* = 0.83 organic, *r* = 0.60 subjective memory complaints). There were no complications from the study and no complaints from the patients concerning the telephone testing. Very occasionally the nurse suspected some cheating. 

This study shows that it is possible to administer the TYM test over the telephone. The telephone TYM test shows very good correlation with an ACE-R test administered by an experienced ACE tester in clinic. Used alone as a cognitive test the tele-TYM shows a sensitivity of 78% and specificity of 69%, with a cut-off set at ≥43. Comparing the patients with organic and subjective memory diagnoses, the tele-TYM score was significantly lower in the organic group (*p* < 0.001, Wilcoxon rank sum test, corrected for multiple comparisons). The results are summarised in Table 4 below.

### 3.2. Longitudinal TYM Studies

Thirty-four patients with Alzheimer’s disease had more than 1 TYM test administered in the Cambridge memory clinic with an interval of between 2 months and 52 months (mean 15 months). There was an average decline in the TYM test of 3.6 points per year.

### 3.3. TYM Passers with a Diagnosis of Amnestic Mild Cognitive Impairment

One of the drawbacks of the TYM test is that it does not reliably detect patients with amnestic mild cognitive impairment or very mild Alzheimer’s disease on total score. To examine this further we selected the 46 patients with a diagnosis of aMCI or mild AD who passed the TYM test and compared them to 157 normal controls. Their results are shown in Table 5.

## 4. Discussion

We looked retrospectively to see whether the pattern of TYM scoring can help distinguish patients with mild amnestic AD who pass the TYM test from controls. The only significant differences between the 2 groups were in recall and help given. Patients with aMCI tended to score comparatively poorly on the second page of the TYM test. One other study [32] has examined the value of the TYM test in the diagnosis of aMCI in a community-based cohort of patients and showed that the TYM test had a specificity of 0.87 and sensitivity of 0.63 in the diagnosis of aMCI.

## 5. Conclusions

The TYM test is a useful short cognitive test in the support of a diagnosis of AD.The TYM test detects a majority of patients with non-Alzheimer’s dementia and unusual presentations of AD.The usefulness of the TYM test has been shown in over 20 published studies in peer-reviewed journals covering different languages, cultures and alphabets.Patients with AD show a decline on the TYM test of 3.6 and 4.1 points/year in two different studies.The TYM test can be used in a general neurology clinic. Patients with AD score significantly lower on the TYM test than patients with no identified neurological cause for their subjective memory problems. Follow-up TYM testing shows that patients with AD decline at an average rate of 4.1 points per year, whilst the patients with subjective memory complaints show a small improvement on average.The TYM test can be used over the telephone with results that correlate with clinic-administered tests, and show a significant difference between patients with organic disease and those with no neurological cause for their subjective memory problems.Patients with amnestic MCI score more poorly than controls on recall and help given on subtests of the TYM.The TYM-MCI is a short cognitive test that can support a clinical diagnosis of mild AD and amnestic MCI.

JMB can be contacted via the website tymtest.com.

## Figures and Tables

**Table 1 diagnostics-09-00116-t001:** Summary of TYM validation studies.

Country	First Author Reference	Numbers Recruited	Setting and Disease	Cut off Used Sensitivity/Specificity or Other Parameter
Japan	Hanyui [7]	159	Memory Clinic AD	Cut off varied Sensitivity 0.96 Specificity 0.88
Japan	Kotuku [8]	334	Memory Clinic AD	Cut off 42 Sensitivity 0.82 Specificity 0.72
United Kingdom	Hancock [3]	224	2 Memory Clinics Dementia	Cut off 30 Sensitivity 0.73 Specificity 0.88
Poland	Szczesniak [9]	225	Memory Clinic AD	Cut off 39 Sensitivity 0.91 Specificity 0.90
Poland	Derkacz [10]	65	Memory Clinic	Cut off 36 Improvement on MMSE (mini-mental state examination)
France	Postel-Vinay [11]	201	Memory Clinic Dementia	Cut off 39 Sensitivity 0.90 Specificity
Greece	Iatraki [12,13]	373	Community and Neurology Clinic Dementia	Cut off varied Sensitivity 0.82 (0.80) Specificity 0.71 (0.77)
South Africa	Van Schalkwyk [14]	100	Primary Care Dementia	Strong correlation with MMSE
Chile Spanish	Munoz-Neira [15]	74	Memory Clinic Dementia	Cut off 39 Specificity 0.93 Sensitivity 0.82
Netherlands	Koekkoek [16]	86	Memory Clinic	AUC = 0.88 Correlated with neuropsychological tests
Netherlands	Van der Zande [17]		Memory Clinic Dementia	Better than MMSE
Turkey	Mavis [18]	395	Memory Clinic Dementia	Cut off 34 Sensitivity 0.97 Specificity 0.96
Argentina Spanish	Serrani [19]	300	Memory Clinic Dementia	Cut off 40 Sensitivity 0.84 Specificity 0.95
Norway	Breitve [20]	33	Memory Clinic	Cut off 42 Specificity 0.84 Sensitivity 1.00 For dementia
Spain	Ferrero-Arias [21]	1049	Neurology Clinic Dementia	Cut off 36 Sensitivity 0.94 Specificity 0.89 For dementia
Iran	Salami [22]	175	Community AD	Cut off 31 Sensitivity 0.9 Specificity 1.0
EgyptArabic	Abd-Al-Atty [23]	206	Hospital Dementia	Cut off 39/40 in well-educated patients only. Sensitivity 0.80 Specificity 0.97
China Mandarin	Xuemei [24]	237	Neurology Clinic AD	Cut off 39.5 Sensitivity 0.95 Specificity 0.95
Hungary	Kolozsvari [25]	50	Community AD	Cut off 35/36 Sensitivity 0.94 Specificity 0.94
Singapore	Dong [26]	90	Memory Clinic AD	Cut off 38 As good as MOCA (Montreal Cognitive Assessment) Better than MMSE AUC 0.96

**Table 2 diagnostics-09-00116-t002:** Characteristics and TYM test results of patients with various diseases in a general neurology clinic. The diagnoses were clinical.

Diagnosis	No.	Age Range Years	Age: Mean and SD Years	TYM Score Range	TYM: Mean and SD
Clinical Alzheimer’s disease	81	48–89	74.7 ± 9.1	0–48	29.6 ± 11.1
Clinical Parkinson’s disease	132	46–90	71.2 ± 9.9	21–50	40.2 ± 7.3
Clinical dementia with Lewy bodies	31	68–84	75.9 ± 5.0	20–46	35.2 ± 7.6
Clinical MCI (Mild Cognitive Impairment)	43	52–86	71.7 ± 9.5	24–49	40.2 ± 7.7
Clinical Epilepsy	30	21–73	58.0 ± 18.1	26–49	39.7 ± 7.2
Subjective memory loss	116	23–88	57.6 ± 14.3	24–50	42.3 ± 5.3

**Table 3 diagnostics-09-00116-t003:** Telephone TYM scores, clinic ACE-R (Addenbrooke’s Cognitive Examination Revised) and clinic MMSE scores in patients with organic dementia or MCI and no organic dementia.

Characteristics	Organic Dementia/MCI	Subjective Memory Complaints
Number in study	38	43
Mean age	69.7	60.5
Telephone TYM score	36.9	43.8
Clinic ACE-R score	71.4	87.6
Clinic MMSE	24.5	27.7

**Table 4 diagnostics-09-00116-t004:** Results of the Telephone TYM study.

Score	Organics Mean	Subjective Memory Complaints Mean	*p* Value Corrected
Tele-TYM overall score	36.9	43.8	<0.001
Orientation	8.9	9.7	0.011
Copy	1.5	1.8	ns
Facts	1.5	2.2	ns
Sums	3.3	3.6	ns
Fluencies	2.2	3.2	0.008
Similarities	3.1	3.5	ns
Naming	4.5	4.6	ns
Visuospatial 1	1.8	2.4	ns
Visuospatial 2	2.9	3.7	0.006
Recall	2.4	4.0	ns
Help	4.7	5.0	0.01

Note: ns = non-significant.

**Table 5 diagnostics-09-00116-t005:** Comparison of TYM and TYM subtest scores in patients with aMCI (amnestic mild cognitive impairment) who passed the TYM test compared to controls.

Characteristic	aMCI	Percentage	Controls	Percentage	*p* = corrected
*n* =	46		157		
Age	68.3		67.5		ns
TYM total/50	45.0	90	46	92	ns
Orientation/10	9.9	99	9.8	98	ns
Copying/2	2.0	98	1.8	90	ns
Facts/3	2.4	81	2.6	87	ns
Sums/4	3.2	81	3.7	93	ns
Fluencies/4	3.4	84	3.4	85	ns
Similarities/4	3.8	94	3.3	83	ns
Naming/5	4.9	99	4.9	98	ns
Visuospatial 1/3	2.8	92	2.8	92	ns
Visuospatial 2/4	3.1	78	3.7	93	ns
Recall/6	3.5	58	5.0	83	<0.001
Help/5	4.7	95	5.0	100	<0.001

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
