# Peer review of "Test Your Memory (TYM) and Test Your Memory for Mild Cognitive Impairment (TYM-MCI): A Review and Update Including Results of Using the TYM Test in a General Neurology Clinic and Using a Telephone Version of the TYM Test"

_diagnostics, 2019, doi:10.3390/diagnostics9030116_

Round 1
Reviewer 1 Report
A useful overview of the TYM test and TYM-MCI. There are a few points which could make this stronger, and some minor typos.
Could the authors please comment on whether they see practice effects if the TYM is administered more than once, and if using a different version alleviates this. Do they have a recommendation on the minimum time which should elapse before test repetition.
Is there any further data available for the AD and PD groups outlined in Table 2, such as disease duration/PD severity? It would be interesting to know if these groups were homogeneous in terms of disease severity. I wonder if baseline disease stage would affect rate of decline on the TYM. Also how much did the rate of change in TYM score vary between individuals - was there a big range or was it fairly consistent within individuals with the same diagnosis?
Editing notes: Table 1 - I assume sensitivity & specificity are for dementia? If so please state.
Line 124 - the sentence appears incomplete.
Table 2 - title text needs editing.
Table 2 - would 'No neurological disease present' be more accurate than 'no neurological cause'?
Lines 193 & 211 - there is a missing qualifier for some of the data - I presume should state 'standard deviation'
Line 239 - should be 'cognition' not 'cognitive'
Author Response
Thank you
We have discussed practice effects and provided a recommended test repetition time
Unfortunately we do not have data on disease severity/duration for the AD and PD groups.
I now comment on individual variability
I have re-written Table 1 including the diagnosis
line 124 completed
Table 2 title edited The phrase subjective memory loss used rather than no neurological cause
Lines 193,211 and 239 completed
Reviewer 2 Report
TYM review
The manuscript combines a review of results of previous TYM-studies with presentation of new results regarding TYM-testing in a non-specialized clinic setting, the Tele-TYM and TYM-testing of amnestic MCI.
The review-section of the paper is relevant, but I have serious reservations regarding the sections where new results are presented.
TYM-testing in a non-specialized clinic setting
Line 173-176: As diagnoses were not definitive and patients were not necessarily followed up, the data are labelled as “pragmatic” and not suited for calculation of diagnostic accuracy statistics. Consequently, the description of the diagnoses should be modified to “possible AD”, “possible PD”, etc.
As the quality of the data seems questionable, the detailed presentation of results on page 6 (line 181-220) is not justified. The information regarding the mean rate of deterioration, however, may be relevant.
Leaving out the presentation of results regarding TYM-testing in a non-specialized clinic setting and leaving out Table 2 from the manuscript would in my opinion improve the paper.
Line 214-215: If the authors choose to retain the information regarding the mean rate of deterioration, they should comment on the small improvement found in the “possible non neurological cause” group. Do they consider it a retest / practice or random effect?
The Telephone TYM
Line 245: “Patients due to be seen in clinic …”. Please specify the type of clinic referred to.
Line 266 and forward: The concept of “organic dementia” vs “no organic dementia” is not clear to this reviewer and possibly somewhat outdated. The “organic cases” are described in line 268-270, but how are “no organic dementia” cases defined?
Line 267: “43 patients were felt not to have an organic cause”. Felt by who? Please clarify the diagnostic procedures applied.
Line 280-281: The results regarding the Tele-TYM are described as “good”, but the diagnostic accuracy statistics (specificity 69%) are not good as more than 30% of the “no organic dementia” cases are misclassified as “organic dementia”.
Other TYM studies
Line 287-296: Again, leaving out the presentation of results regarding TYM-testing of 245 patients from the Cambridge Memory clinic and leaving out Table 5 from the manuscript would in my opinion improve the paper.
The results regarding the average rate of decline, however, may be relevant.
TYM Passers with aMCI
Line 312-313: The aMCI group performed slightly worse than controls on Visuospatial 2/4, but according to Table 6, the difference was not significant. Consider elaborating only on significant findings.
Line 313-314: The aMCI group performed (non-significantly) better than controls on Similarities – possibly due to education bias. This hypothesis is not substantiated by the data. Please report facts regarding the educational level of the aMCI group versus controls or consider leaving out the hypothesis.
The authors claim that the aMCI group also performed better than controls on Fluencies, but according to Table 6, the two groups had identical scores (3.4). Consider revising.
Line 319-320: The results regarding aMCI are described as “good”, but the diagnostic accuracy statistics (sensitivity 63%) are not good as more than 35% of aMCI cases are not identified.
Conclusions
Conclusions no 5, 7 and 8 are, in the opinion of this reviewer, not substantiated by the results presented in the paper. Please consider leaving out these conclusions.
Author Response
Thank you.
TYM in non specialized clinic setting
lines 173-6 I have changed the diagnostic labels to clinical AD and clinical PD.
I have now cut out virtually all the analysis of this data just leaving the significant difference between patients with clinical AD and subjective memory complaints.
I feel that provide we make the limitations of the data clear (and I think we do) then how the TYM performs in a real life NHS General Neurology clinic is of interest.
We do include the mean rate of deterioration and comment now on this
Telephone TYM
clinic type is specified
line 266 I agree how to label these patients is controversial, we have stated how this was defined and use the alternative label of subjective memory complaints line 267 I have now made this clear
line 280 I have deleted the word good
Other TYM studies
I have now deleted this section and table 5 whilst retaining the data on rate of decline
TYM passers with aMCI
I have now cut much of this section and only discuss the significant findings as suggested.
line 319. I have cut the word good
Conclusions
I have now modified conclusions 5,7 and 8
Round 2
Reviewer 2 Report
Thank you for the revision. I have no further comments.